# Burden and predictors of heart failure treatment outcomes in Ethiopia: A systematic review and meta-analysis protocol

**Firomsa Bekele**[1]*, **Lalise Tafese**[2], **Bayisa Garbessa**[3], **Shimalis Tadasa**[4], **Ginenus Fekadu**[1,5]

**1** Department of Pharmacy, Institute of Health Science, Wallaga University, Nekemte, Ethiopia,
**2** Department of Health informatics, College of Health Science, Mattu University, Mattu, Ethiopia,
**3** Department of Anesthesia, College of Medicine and Health Science, Dire Dawa University, Dire Dawa, Ethiopia, **4** Department of Medicine, College of Health Science, Mizan Tepi University, Mizan, Ethiopia,
**5** School of Pharmacy, Faculty of Medicine, The Chinese University of Hong Kong, Shatin, N.T, Hong Kong, China

* firomsabekele21@gmail.com

**Data Availability Statement:** The data are only available upon request. The data would be guarded carefully by our third party data for the only purpose of this scientific study. Participants were

## Abstract

### Background

Heart failure is an important global health problem which is associated with high mortality. Uncontrolled heart failure leads to hospitalization and reduction in quality of life. Therefore, the study aimed to assess the treatment outcome such as improved, death, hospitalization, and self-discharges without improvement and associated factors in heart failure patients admitted to south western Ethiopian hospitals.

### Methods

We will use databases such as PubMed, Science Direct, HINARI, Scopus and Google Scholar. The final systematic review and meta-analysis will contain papers that fulfill the eligible criteria. A systematic data extraction check list will be used to extract the data, and STATA version 14 will be used for the analysis. Heterogeneity is evaluated using the $I^2$ tests and the Cochrane Q test statistic. To examine publication bias, a funnel plot, Egger's weighted regression, and Begg's test are utilized. The sensitivity analysis and subgroup analysis will be done for studies having heterogeneity. The Joanna Briggs institute meta-analysis of statistics assessment and review instrument (JBI- MAStARI) will be used for quality assessment.

### Discussion

This protocol is expected to provide adequate evidence on the burden of poor heart failure treatment outcome that includes self-discharge, developing complication and finally leads to death in acute and chronic heart failure patients in Ethiopia. Furthermore, to enrich our estimation, we also intended to assess the associated factors of poor treatment outcome. Therefore, our review will call for government and non-government interventions in reducing the mortality associated with heart failure.

not signed consent for data publicly. For all these reasons and following the indications of the research review committee of College of Health Sciences, Mettu University, the authors must not upload the dataset to a stable, public repository. Interested, qualified researchers can access the data by requesting our third party, Mattu University (mattuniversity@meu.edu.et).

**Funding:** The author(s) received no specific funding for this work.

**Competing interests:** The authors have declared that no competing interests exist.

**Abbreviations:** OR, Odds Ratio; CHF, Congestive Heart failure; CI, Confidence Interval; LMIC, low- and middle-income countries; PRISMA, Preferred Reporting Items for Systematic Reviews and Meta-analyses.

## Introduction

Heart failure (HF) is a complex clinical syndrome characterized by the reduced ability of the heart to pump and/or fill with blood. It can also be defined as an inadequate cardiac output to meet metabolic demands [1]. The complex clinical syndrome of HF is associated with a wide spectrum of left ventricular functional abnormalities that is manifested as dyspnea, fatigue, poor exercise tolerance, and fluid retention [2, 3].

Globally, the magnitude of heart failure is 64.34 million [4]. HF is a well-recognized public health problem representing a significant burden for patients and healthcare systems in developed countries [5]. It is a major public health issue, with a prevalence of 6.5 million in the United States (U.S) [6]. The prevalence of HF in the U.S is projected to increase by 46% between 2012 and 2030 (3). HF is a growing problem in Africa, particularly in sub-Saharan Africa [7]. A study conducted in Ethiopia indicated that HF is one of the most common cardiovascular diseases which counted 23.9% [8].

In Ethiopia, different heart failure treatment outcomes such as improved, death, hospitalization, and self-discharge without improvement were reported. High in-hospital mortality (17.2%) was observed among acute heart failure patients admitted to a Tikur Anbessa Specialized Hospital, Addis Ababa [9], Debre Markos Comprehensive Specialized Hospital (12.7%) [10], and University of Gondar Comprehensive Specialized Hospital(10.6%) [11]. In Jimma Medical Center, a total of 51 (21.1%) patients were hospitalized, and 58 (24.0%) of patients had worsened clinical states [12]. The limitation of the previous study was the single setting of the study that cannot be generalized to other settings. The various reports of heart failure mortality and hospitalization necessitate the systematic review and meta-analysis study across Ethiopia.

The good self-care practices among heart failure a patient is viewed as the cornerstone of heart failure (HF) therapy and entails essential practices that have been shown to improve HF clinical outcomes. These include appropriate adherence to prescribed medication, sodium diet restriction, physical activity, reducing liquid intake, and weight reduction [13]. Besides to non-pharmacological management of heart failure, Angiotensin-converting enzyme inhibitors, beta-blockers, and mineralocorticoid receptor antagonists are medications that can be used to treat heart failure (HF) and reduce mortality and hospitalization [12, 14].

Coordinated discharge planning and the establishment of a properly structured follow-up treatment are two strategies that have been created to lower mortality and morbidity during the sensitive post-discharge period [15]. However, HF hospitalizations and readmissions are frequent, and residual mortality is still high in developed nations [5, 16–18]. Within the first 30 days of discharge, 20% of HF patients were readmitted to the hospital [15]. Heart failure treatment outcomes that includes improved, death, hospitalization, and self-discharge without improvement were poor, according to several studies conducted in Ethiopia [9, 12, 19].

It has been shown the presence of co-morbidity, drug therapy problem, smoking cigarette are the risk factors that are associated with poor treatment outcomes such as death, hospitalization, and self-discharges without improvement [9, 20, 21]. Despite a variable reports of magnitude and predictors of heart failure treatment outcomes (improved, death, hospitalization, and self-discharge without improvement) in Ethiopia, there was no systematic review and meta-analysis conducted and the purpose of this paper is to develop a protocol to summarize the recent findings on factors related to poor treatment outcome such as death, hospitalization, and self-discharges without improvement in order to provide an appropriate intervention.

The study will identify the predictors of mortality among heart failure patients. The finding will help in optimizing patient's heart failure therapy by providing a recommendation for hospital administrators and health care organization to implement Clinical Pharmacy services in health care policy of Ethiopia. The Clinical Pharmacy services will also reduce the treatment burden, hospitalization and untoward effects of heart failure medications.

The study will assist policy makers in improving the evidence for planning various types of interventions in the specific context of decreasing mortality from heart failure. Besides, it will help the government in developing the standard treatment guidelines and evaluating the effect of their program on heart failure patients. The review will help in the clinical practice by providing appropriate treatments and addressing factors associated with poor heart failure outcome such as death, hospitalization, and self-discharges without improvement. Finally, our study will be used as a benchmark for future researchers conducting the interventional studies to reduce heart failure hospitalization and mortality.

## Study objectives

**General objectives.**   The aim of this systematic review and meta-analysis is to estimate the pooled magnitude of heart failure treatment outcome and associated factors

**Specific objectives.**   To assess the burden of heart failure treatment outcome in Ethiopia. To assess the risk factors of heart failure treatment outcome in Ethiopia.

## Research questions

**General questions.**   This protocol aims to answering the questions: what is the burden of heart failure treatment outcome in Ethiopia, as well as what determines the poor treatment outcome?

**Specific questions.**   What is the burden of poor heart failure treatment outcome in Ethiopia?

What are the associated factors of heart failure treatment outcome?

## Materials and methods

### Study design

The protocol of Preferred Reporting Items for Systematic Reviews and Meta-analyses (PRISMA-P) checklist will be used to undertake this systematic review and meta-analysis [22]. The Review protocol was registered on PROSPERO CRD 42023437397. The PRISMA flow diagram will be used [Fig 1].

### Quality assessment and risk of bias

The Joanna Briggs institute meta-analysis of statistics assessment and review instrument (JBI-MAStARI) will be used for quality assessment [23]. The criteria used for assessing the quality of included studies are as follows; are the criteria for inclusion in the sample clearly defined?, are the study subjects and the setting described in detail?, is the exposure measured in a valid and reliable way?, are objective, standard criteria used for measurement of the condition?, are confounding factors identified?, are strategies to deal with confounding factors stated?, are the outcomes measured in a valid and reliable way?, and is appropriate statistical analysis used?.

Two authors will assess the risk of bias among the studies. The disagreement between the authors will be solved by discussion. The type of bias will be identified for all studies

**Outcome measurement.**   There are two main outcomes. The primary outcome of interest is the prevalence of poor heart failure treatment outcome, which will be estimated as the total

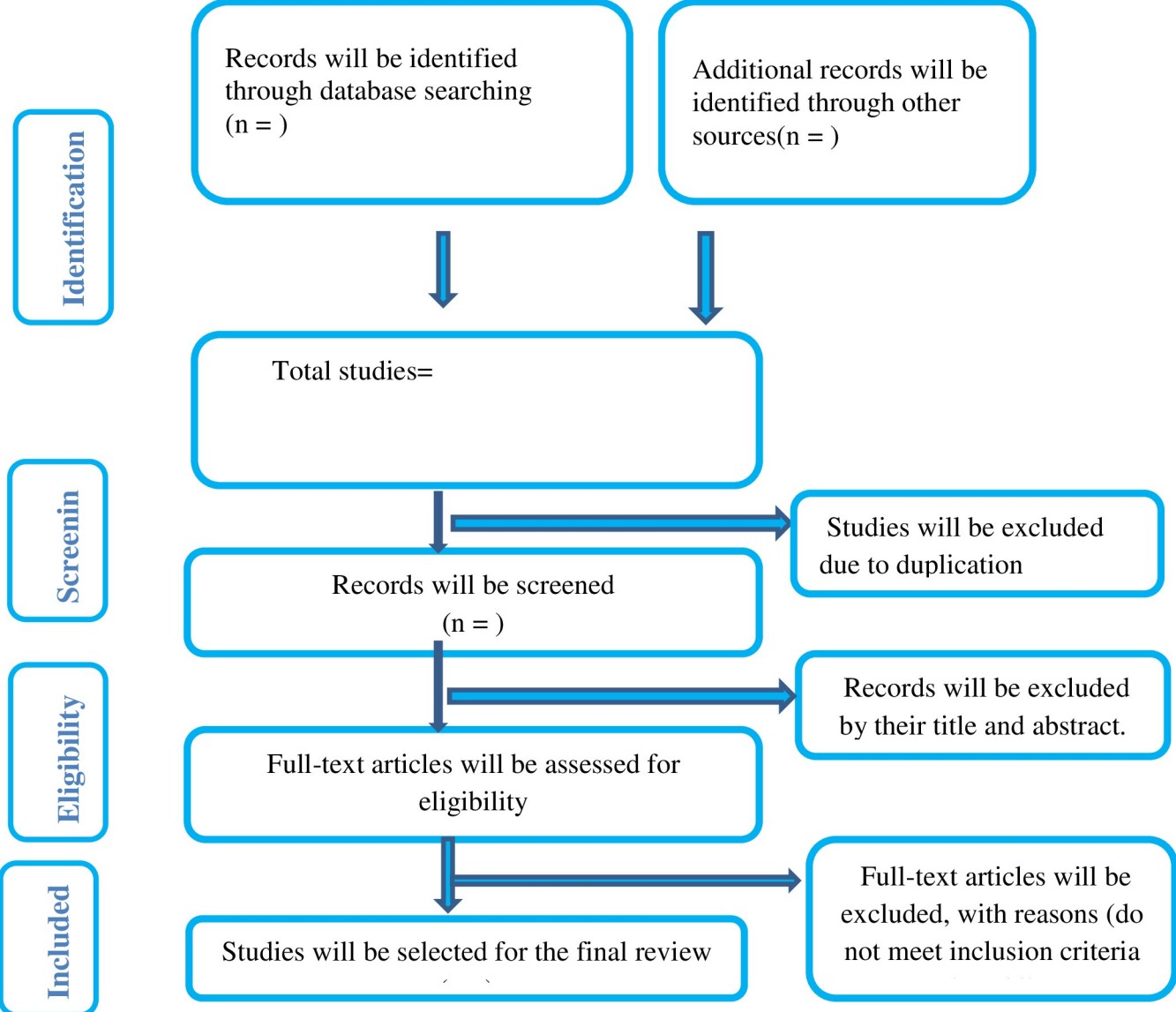

**Fig 1. Flow chart of the systematic research and study selection process.**

number of patients having poor treatment outcome divided by the total number of heart failure patients multiplied by 100. The second outcome is identifying factors associated with poor treatment outcome, which are determined using the odds ratio (OR) and calculated based on binary outcomes from the included primary studies. Independent variables that have association with the primary outcome (heart failure treatment outcome) will be included in the meta-analysis in order to identify the association between heart failure treatment outcome and independent variables that includes socio-demographic and clinical characteristics. The pooled finding of the meta-analysis will be identified to predict the final pooled association. This will be done on Microsoft excel heet as listing as authors name, factors associated with heart failure treatment outcome, odds ratio, lower class limit, upper class limit, log of lower class limit, log of upper class limit, log of odds ratio and selog of odds ratio.

**Operational definitions.** *Treatment outcome*. The achievement of a specified end result like improvement and/or mortality [9].

*Poor outcome*. Hospital mortality secondary to congestive heart failure (CHF), self-discharges, and presents with complications [9].

*Good outcome*. Patients that was improved at discharge [9].

## Eligibility criteria

The findings published related to magnitude and predictors of heart failure treatment outcome in Ethiopia having all primary outcome and full texts available will be included. The articles with unknown primary outcomes, systematic reviews and meta-analysis studies, not peer reviewed and commentary to editors will not be eligible. The review will use the CoCoPop (condition, Context, and Population) framework to assess the eligibility of the studies.

The study Population (POP) is the adult heart failure patients greater than 18 years having a co-morbidity of renal insufficiency, stroke, diabetes mellitus, chronic obstructive pulmonary disease, hypertension, and anemia. The Condition (CO) is heart failure treatment outcome that includes improved, death, hospitalization, and self-discharge without improvement, and the context (CO) studies conducted in Ethiopia.

## Searching strategy

The time period used to conduct search of the literature review will be on September 21, 2024. The primary searching data bases will includes PubMed, HINARI, Science direct, Scopus and Google scholar. The primary data on magnitude and associated factors of heart failure treatment outcome will be retrieved. The search term will include the MESH terms, Boolean operators, free text, key words and gray literature searches. The MESH term for the database is ((heart failure treatment outcome) OR (prevalence)) mortality OR (associated factors) AND (Ethiopia).

## Data collection process, items and extraction

Two authors namely FB and LT will be involved in collecting different literatures. Reference management software (endnote version X7.2) will be used to combine search results from databases and to remove duplicate articles. Data will be extracted by three data extractors (ST, GF and BG) using a standardized data extraction checklist on Microsoft excel. For the first outcome (magnitude), the data extraction checklist includes author name, year of publication, region, study design, sample size and number of participants with the outcome. For the second outcome (associated factors), data will be extracted in a format of two by two tables, and then the log OR for each factor will be calculated based on the findings of the original studies. Discrepancies between two independent reviewers will be resolved by involving a third reviewer (FB) after discussion for possible consensus. LT and GF will oversee the overall process of data extraction and synthesis.

**Ethical approval and consent to participate.** No ethical approval will be needed because data from previous published studies in which informed consent was obtained by primary investigators will be retrieved and analyzed. No human subject participants will be involved. The participant recruitment dates and/or date on which medical records were accessed were mentioned by previous studies.

**Data analysis and synthesis.** Data will be exported to STATA V. 14 to calculate the pooled effect size with 95% CIs. To check heterogeneity among the studies, the Cochran $Q$ test (chi-squared statistic) and $I^2$ statistic on forest plots will be computed. Cochran's $Q$ statistical heterogeneity test will be considered statistically significant at $P \leq 0.05$. $I^2$ statistics range from

0 to 100% and $I^2$ statistic values of 0, 25, 50, and 75% are considered as no, low, moderate, and high degrees of heterogeneity, respectively. The subgroup analysis and sensitivity analysis will be done if the heterogeneity is reported. A random effect model was used if heterogeneity is present and whereas a fixed effect model is used if the study variation is not occurred. A funnel plot will be used to assess publication bias. Asymmetry of the funnel plot is an indicator of publication bias. Besides, Egger's weighted regression and Begg's test will be used to check publication bias. Statistical significance of publication bias will be declared at a P-value of less than 0.05.

## Discussion

Heart failure, which affects around 26 million people globally and is linked to high mortality, is a significant global health issue. The industrialized world provides the majority of the information on heart failure patient outcomes, while underdeveloped nations like Ethiopia provide significantly less data. Based on the little information available, low- and middle-income countries' (LMIC) heart failure patient mortality may be higher than in high-income nations. As a result, both cardiac variables such as decreased ejection fraction, prolonged hospital stay, advanced stage of heart failure, higher class of heart failure and non-cardiac variables that includes being an elder, being a male, the presence of medication related problem, drinking alcohol and the presence of co-morbidity continued to cause significant geographical variations in heart failure patient mortality. The infrastructure, quality, and accessibility of healthcare as well as environmental and genetic factors may all contribute to regional disparities in mortality.

According to the authors' knowledge, there hasn't been a review published yet on determining how Ethiopian patients with heart failure are being treated thus far. The study will aim to identify the burden and predictors of poor heart failure outcomes that includes mortality, self-discharge without improvement of the disease in poor resource settings. The finding will update the evidence based decisions among heart failure patients in Ethiopia.

Both published and unpublished research will be considered in the investigation in order to better understand the pooled prevalence of heart failure treatment result and its associated factors. Hence, the systematic reviews and meta-analysis study is needed to summarize the outcome of heart failure in Ethiopia. Furthermore, to enrich our estimation, we will also intended to assess the prevalence of poor treatment outcome that includes self-discharge, developing complication and finally leads to death in acute and chronic heart failure patients. Besides, we aim to assess the predictors of poor treatment outcome to early tackle the worsening situations. Therefore, our review will call for government and non-government interventions in reducing the mortality associated with heart failure. The study will assist policy makers in improving the evidence for planning various types of interventions in the specific context of decreasing mortality from heart failure. Besides, it will help the government in developing the standard treatment guidelines and evaluating the effect of their program on heart failure patients. The review will help in the clinical practice by providing appropriate treatments and addressing factors associated with poor heart failure outcome such as mortality. Finally, our study will be used as a benchmark for future researchers conducting the interventional studies to reduce heart failure hospitalization and mortality.

Undoubtedly, the result of our review will be disseminated to the stakeholders through research conferences, and workshops. Furthermore, the finding will be published in highly ranked journals. Including studies published only in the English language, which might impose bias on the outcome, and the lack of free access to some databases like EMBASE is the

limitation of the study. Therefore, the finding of the study should carefully generalize to outsides of the study area.

## Conclusion

The finding of our review will provide the insight for the burden of heart failure in poor resource country. The result might provide a baseline for developing the interventions and evaluating the program to combat burden of the disease. It will also help identify risk factors of mortality secondary to heart failure. Finally it will provide a clue on the importance of clinical pharmacist in monitoring drug therapy of heart failure patients and designing standard treatment guidelines for hospitalized heart failure patients.

## Supporting information

**S1 Checklist. PRISMA-P (Preferred Reporting Items for Systematic review and Meta-Analysis Protocols) 2015 checklist.**
(DOC)

## Author Contributions

**Conceptualization:** Firomsa Bekele, Lalise Tafese, Bayisa Garbessa, Ginenus Fekadu.

**Data curation:** Bayisa Garbessa, Ginenus Fekadu.

**Formal analysis:** Firomsa Bekele, Bayisa Garbessa, Ginenus Fekadu.

**Investigation:** Firomsa Bekele, Shimalis Tadasa.

**Methodology:** Firomsa Bekele, Lalise Tafese, Bayisa Garbessa.

**Resources:** Shimalis Tadasa.

**Supervision:** Lalise Tafese.

**Writing – review & editing:** Shimalis Tadasa.

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
