## [Decision Letter · Decision Letter 0]

22 Aug 2023

PONE-D-23-18181Burden and predictors of heart failure treatment outcomes in Ethiopia: A systematic review and meta-analysis protocolPLOS ONE

Dear Dr. Bekele,

Thank you for submitting your manuscript to PLOS ONE. After careful consideration, we feel that it has merit but does not fully meet PLOS ONE’s publication criteria as it currently stands. Therefore, we invite you to submit a revised version of the manuscript that addresses the points raised during the review process.

We look forward to receiving your revised manuscript.

Kind regards,

Tariq Jamal Siddiqi

Academic Editor

PLOS ONE

Journal Requirements:

3. Please include a caption for figure 1.

Reviewers' comments:

Reviewer's Responses to Questions

**Comments to the Author**

1. Does the manuscript provide a valid rationale for the proposed study, with clearly identified and justified research questions?

Reviewer #1: Yes

Reviewer #2: Yes

2. Is the protocol technically sound and planned in a manner that will lead to a meaningful outcome and allow testing the stated hypotheses?

Reviewer #1: Yes

Reviewer #2: Yes

3. Is the methodology feasible and described in sufficient detail to allow the work to be replicable?

Reviewer #1: Yes

Reviewer #2: Yes

4. Have the authors described where all data underlying the findings will be made available when the study is complete?

Reviewer #1: Yes

Reviewer #2: Yes

5. Is the manuscript presented in an intelligible fashion and written in standard English?

Reviewer #1: Yes

Reviewer #2: Yes

6. Review Comments to the Author

You may also provide optional suggestions and comments to authors that they might find helpful in planning their study.

Reviewer #1: Thank you for submitting your article for consideration to Plos One. The meta-analysis protocol is well-written and detailed. Some minor additions will help strengthen the protocol.

1) It would be beneficial to provide details on how secondary outcomes will be handled, or why they might not be applicable to this study.

2) While the discussion is detailed and well-thought-out, it may be helpful to give more emphasis on the impact of this study on policy-making, clinical practice, and future research in Ethiopia.

Reviewer #2: In their study protocol on “Burden and predictors of heart failure treatment outcome in Ethiopia: A systematic review and meta-analysis protocol,” the authors present a crucial investigation that addresses a significant global health issue. Heart Failure affects millions worldwide, and optimizing treatment outcomes in this population is of paramount importance for their overall well-being and healthcare management in Ethiopia. However, to further enhance the impact and robustness of this study protocol the authors should consider addressing the following points.

1. Pages 8-9, Introduction Section: The introduction efficiently presents the topic of the study. However, the authors should consider providing a comprehensive review of the existing literature to further enhance the introduction. For instance, the authors could include more specific details from relevant studies in Ethiopia related to heart failure outcomes and address the strengths and weaknesses of previous research. By doing so, the authors can better set the context for their study and emphasize its significance.

2. Pages 8-9, Introduction Section: The authors could also further explain the potential impact of the study on patient care and healthcare interventions in Ethiopia by explaining how the findings may improve clinical practice and policies. This will strengthen the study's credibility and create a sense of importance and urgency in addressing heart failure treatment outcomes.

3. Pages 8-9, Introduction Section: The term “treatment outcome” has been used throughout the introduction, but it would be helpful if the authors could further elaborate on this and explain in the introduction section what treatment outcome entails in the context of heart failure in Ethiopia. This will help improve the clarity and understanding of the reader.

4. Pages 9-11, Methodology Section: The authors have provided a very well-structured and comprehensive methodology. However, they should consider clarifying and providing more specific details about the Condition (CO) and Population (POP) of the study. For example, they could specify the age ranges and comorbidities of the population and explain the type and severity of heart failure treatment outcomes considered. This will help enhance the robustness and reliability of the study.

5. Pages 9-11, Methodology Section: While the methodology mentions using The Joanna Briggs Institute meta-analysis of statistics assessment and review instrument (JBIMAStARI) for quality assessment. It would be helpful if the authors could provide a brief explanation of the key elements assessed in (JBIMAStARI), as doing so will improve the reliability and transparency of the study.

6. Page 12, Discussion Section: The discussion effectively mentions the potential geographical variation in heart failure patient mortality due to cardiac and non-cardiac variables. However, to further strengthen the discussion the authors should consider providing specific examples or explanations of these variables. This will help improve the readers’ understanding.

7. Page 12, Discussion Section: While the authors acknowledge the absence of published reviews on Ethiopian patients with heart failure, the discussion could be improved by highlighting the unique nature of this study's objectives and its potential to guide evidence-based interventions. This will help underscore the significance and relevance of the research in advancing heart failure management and care in Ethiopian patients.

8. Page 12, Discussion Section: This section briefly mentions language bias and lack of access to some databases as limitations. To improve the study protocol, it would be helpful to elaborate on the potential implications of these limitations on the generalizability of the study's findings. By doing so the authors can bolster the overall rigor and reliability of the study.

9. Page 12, Discussion Section: Please consider revising the phrase “including studies published only in English language that might impose bias on the outcome and lack of free assess of some database like EMBASE.” To “including studies published only in the English language, which might impose bias on the outcome, and the lack of free access to some databases like EMBASE.” This will help improve the clarity and readability of the text.

7. PLOS authors have the option to publish the peer review history of their article (what does this mean?). If published, this will include your full peer review and any attached files.

Reviewer #1: **Yes: **Mahammed Z Khan suheb

Reviewer #2: No

---

## [Author Response · Author response to Decision Letter 0]

29 Aug 2023

Tariq Jamal Siddiqi

Academic Editor of PLOS ONE

Dear Editor of the manuscript PONE-D-23-18181 entitled “Burden and predictors of heart failure treatment outcomes in Ethiopia: A systematic review and meta-analysis protocol" submitted to PLOS ONE. Thanks for your time and consideration in editing and reviewing the manuscript. We have carefully read your comments and corrected inline of your comments and suggestions. All comments raised were edited and incorporated in the revised manuscript. Here are the responses and elaborations for the comments from the reviewers!

REVIEWERS COMMENT

Reviewer 1: 

Reviewer comment: It would be beneficial to provide details on how secondary outcomes will be handled, or why they might not be applicable to this study.

Author response: We have mentioned the secondary outcomes and clearly stated the procedures to handle it in revised manuscript under Outcome measurement section of revised manuscript

Reviewer comment: While the discussion is detailed and well-thought-out, it may be helpful to give more emphasis on the impact of this study on policy-making, clinical practice, and future research in Ethiopia.

Author response: The impact of the study was discussed in revised manuscript

Reviewer 2:

Reviewer comment: Pages 8-9, Introduction Section: The introduction efficiently presents the topic of the study. However, the authors should consider providing a comprehensive review of the existing literature to further enhance the introduction. For instance, the authors could include more specific details from relevant studies in Ethiopia related to heart failure outcomes and address the strengths and weaknesses of previous research. By doing so, the authors can better set the context for their study and emphasize its significance.

Author response: We have included additional the important literatures related to heart failure outcomes under introduction section of revised manuscript. The strengths and weaknesses of previous research were mentioned.

Reviewer comment: Pages 8-9, Introduction Section: The authors could also further explain the potential impact of the study on patient care and healthcare interventions in Ethiopia by explaining how the findings may improve clinical practice and policies. This will strengthen the study's credibility and create a sense of importance and urgency in addressing heart failure treatment outcomes.

 Author response: The significance/impact of the study was described in revised manuscript

Reviewer comment: Pages 8-9, Introduction Section: The term “treatment outcome” has been used throughout the introduction, but it would be helpful if the authors could further elaborate on this and explain in the introduction section what treatment outcome entails in the context of heart failure in Ethiopia. This will help improve the clarity and understanding of the reader.

 Author response: We have elaborated it in revised manuscript as heart failure treatment outcomes includes improved, death, hospitalization, and self-discharge without improvement 

Reviewer comment: Pages 9-11, Methodology Section: The authors have provided a very well-structured and comprehensive methodology. However, they should consider clarifying and providing more specific details about the Condition (CO) and Population (POP) of the study. For example, they could specify the age ranges and comorbidities of the population and explain the type and severity of heart failure treatment outcomes considered. This will help enhance the robustness and reliability of the study.

Author response: We have provided more details of Condition and Population in revised manuscript

Reviewer comment: Pages 9-11, Methodology Section: While the methodology mentions using The Joanna Briggs Institute meta-analysis of statistics assessment and review instrument (JBIMAStARI) for quality assessment. It would be helpful if the authors could provide a brief explanation of the key elements assessed in (JBIMAStARI), as doing so will improve the reliability and transparency of the study.

 Author response: The key elements of JBIMAStARI was described under the Quality assessment and risk of bias section of revised manuscript.

Reviewer comment: Page 12, Discussion Section: The discussion effectively mentions the potential geographical variation in heart failure patient mortality due to cardiac and non-cardiac variables. However, to further strengthen the discussion the authors should consider providing specific examples or explanations of these variables. This will help improve the readers’ understanding.

 Author response: We have mentions the specific examples of cardiac and non-cardiac variables in revised manuscript as the cardiac variables includes decreased ejection fraction, prolonged hospital stay, advanced stage of heart failure, higher class of heart failure and non-cardiac variables includes being an elder, being a male, the presence of medication related problem, drinking alcohol and the presence of co-morbidity.

Reviewer comment: Page 12, Discussion Section: While the authors acknowledge the absence of published reviews on Ethiopian patients with heart failure, the discussion could be improved by highlighting the unique nature of this study's objectives and its potential to guide evidence-based interventions. This will help underscore the significance and relevance of the research in advancing heart failure management and care in Ethiopian patients.

 Author response: We have added the study objectives and impact of the study on health care interventions in the revised manuscript.

Reviewer comment: Page 12, Discussion Section: This section briefly mentions language bias and lack of access to some databases as limitations. To improve the study protocol, it would be helpful to elaborate on the potential implications of these limitations on the generalizability of the study's findings. By doing so the authors can bolster the overall rigor and reliability of the study.

Author response: We have added the potential implications of limitations on the generalizability of the study's findings in revised manuscript

Reviewer comment: Page 12, Discussion Section: Please consider revising the phrase “including studies published only in English language that might impose bias on the outcome and lack of free assess of some database like EMBASE.” To “including studies published only in the English language, which might impose bias on the outcome, and the lack of free access to some databases like EMBASE.” This will help improve the clarity and readability of the text.

Author response: We have edited as per your comment

Thanks for your time and consideration,

 Regards!

---

## [Editor Report · Decision Letter 1]

4 Sep 2023

Burden and predictors of heart failure treatment outcomes in Ethiopia: A systematic review and meta-analysis protocol

PONE-D-23-18181R1

Dear Dr. Bekele,

We’re pleased to inform you that your manuscript has been judged scientifically suitable for publication and will be formally accepted for publication once it meets all outstanding technical requirements.

Kind regards,

Tariq Jamal Siddiqi

Academic Editor

PLOS ONE
---

## [Editor Report · Acceptance letter]

13 Dec 2023

PONE-D-23-18181R1 

PLOS ONE

Dear Dr. Bekele Negera, 

I'm pleased to inform you that your manuscript has been deemed suitable for publication in PLOS ONE. Congratulations! Your manuscript is now being handed over to our production team.

Kind regards, 

on behalf of

Dr. Tariq Jamal Siddiqi 

Academic Editor

PLOS ONE